# Smokers' strategies across social grades to minimise the cost of smoking in a period with annual tax increases: evidence from a national survey in England

Mirte AG Kuipers,[1,2] Timea Partos,[3] Ann McNeill,[3] Emma Beard,[2,4] Anna B Gilmore,[5] Robert West,[2] Jamie Brown[2,4]

For numbered affiliations see end of article.

**Correspondence to**
Dr Mirte AG Kuipers;
m.a.kuipers@amsterdamumc.nl

## ABSTRACT

**Objectives** To assess associations between smokers' strategies to minimise how much their smoking costs and cost of smoking among smokers across three social grades during a period of annual tax increases in England.

**Design** Repeat cross-sectional.

**Setting** England, May 2012–December 2016.

**Participants** 16 967 adult smokers in 56 monthly surveys with nationally representative samples.

**Measures and analysis** Weighted generalised additive models assessed associations between four cost-minimising strategies (factory-made and roll-your-own (RYO) cigarette consumption levels, illicit and cross-border purchases) and cost of smoking (£/week). We adjusted for inflation rate, age, gender and secular and seasonal trends.

**Results** Cost of smoking did not increase above the rate of inflation. Factory-made cigarette consumption decreased, while proportion of RYO and, to a much lesser extent, illicit and cross-border purchases increased. These trends were only evident in lowest social grade. Cost of smoking was 12.99% lower with consumption of 10 fewer factory-made cigarettes (95% CI −13.18 to −12.80) and 5.86% lower with consumption of 10 fewer RYO cigarettes (95% CI −5.66 to −6.06). Consumption levels accounted for 60% of variance in cost. Cross-border and illicit tobacco purchases were associated with 9.64% (95% CI −12.94 to −6.33) and 9.47% (95% CI −12.74 to −6.20) lower costs, respectively, but due to low prevalence, accounted for only 0.2% of variation. Associations were similar across social grades, although weaker for illicit and cross-border purchases and stronger for consumption in higher social grades compared with lower social grades.

**Conclusion** During a period of annual tax increases, the weekly cost of smoking did not increase above inflation. Cost-minimising strategies increased, especially among more disadvantaged smokers. Reducing cigarette consumption and switching to RYO tobacco explained a large part of cost variation, while use of illicit and cross-border purchasing played only a minor role.

## Strengths and limitations of this study

► This study used monthly data on a 4.5-year period from a large nationally representative sample of smokers across social grades.

► Data allowed comparison of the contribution of multiple cost minimising strategies to the cost of smoking.

► Illicit and cross-border purchases were measured as any purchases over the past 6 months, and the variables may therefore not reflect the frequency of the use of these sources.

► The data did not contain information on usual brand of cigarettes, and we therefore did not take brand switching into account.

► We cannot rule out selective quitting, which may have affected the observed trend in cost of smoking if smokers who spend less are more likely to quit than those who spend more.

## INTRODUCTION

Increasing taxes on tobacco is considered among the most effective ways of reducing smoking prevalence.[1] In line with economic theory,[2] the resulting increased costs of smoking may provoke quit attempts, reduce smoking consumption and deter uptake.[3 4] Notably, tax increases are among the few interventions to show greater effectiveness in lower, compared with higher, socioeconomic strata,[5] which is important given the association between smoking and disadvantage in England[6] and other countries with a mature smoking epidemic.[7] The effectiveness of tax increases may, however, be diminished by the increased availability of cheap tobacco[8 9] and cost-minimising strategies that smokers develop in response.[10 11]

Studies across high-income countries estimate that between half and three quarters of

all smokers apply cost-minimising strategies[10–14] such as cutting down cigarette consumption, switching to a lower priced brand, switching to roll-your-own (RYO) tobacco or evading or avoiding tobacco taxation by changing the source of purchase to illicit or cross-border/duty free sources, respectively (collectively known as non-UK duty-paid).[8 10–13] US smokers who use one or more cost-minimising strategy have been shown to significantly reduce their costs, on average by 22%.[14 15] Moreover, the use of cost-minimising strategies shows social patterning[16] with smokers of lower socioeconomic status (SES) being more likely to smoke RYO cigarettes and use cheaper factory-made cigarette brands,[12 17] and smokers of higher SES being more likely to purchase untaxed tobacco products.[12 18] The relative importance of different strategies, however, remains to be quantified, as well as the extent to which strategies differentially affect the cost of smoking among different socioeconomic groups.

The current study was set in the UK, where there has been a marked decrease in smoking prevalence (from 19.3% in 2012 to 15.5% in 2016[19]) and where some use of cost-minimising strategies has been demonstrated using data from the International Tobacco Control policy evaluation project.[8 11 12 18] The UK has among the highest tobacco tax rates worldwide.[20] In 2012–2016, taxes on all tobacco products increased nationwide in March of each year, by 5% above inflation in 2012 and 2% above inflation in 2013–2015. In 2016, a differential tax of 5% above inflation for RYO tobacco and 2% above inflation for factory-made cigarettes was applied.[21] Between 2002 and 2014, the proportion of UK smokers who used at least some RYO tobacco increased substantially, from 30% to 45%.[8] With RYO prices (per stick) being less than half of those of factory-made cigarettes,[8 9] switching to RYO seems effective in lowering the cost of smoking. Between 2002 and 2014, use of tobacco from non-UK duty-paid sources remained consistent or even decreased,[8 22] but increased slightly after 2015.[22] The extent to which the use of non-UK duty-paid tobacco contributes to mitigating the costs of smoking, and its relative importance compared with RYO tobacco, has not been previously established, especially across different social grade levels.

This study examined the extent to which the use of cost-minimising strategies allowed smokers from different social grades in England to minimise the actual cost of smoking in 2012–2016. We used data from the Smoking Toolkit Study (STS), which is a repeated cross-sectional monthly survey of the English population. The use of the STS allowed for the study of much more detailed trends and comprehensive measures of illicit tobacco use, in recent nationally representative data.

Specifically, the aims of this study were:

1. to describe trends in the cost of smoking (smokers' self-reported weekly spend on tobacco) and use of cost-minimising strategies between 2012 and 2016 overall and by social grade;
2. to assess in the general smoking population and across social grades, the association of cost of smoking with:

3. the number of cigarettes smoked per week, both factory-made and RYO;
4. and the purchase of non-UK duty-paid tobacco by means of purchase from either illicit or cross-border sources.

## METHODS

### Data and study population

Data were collected as part of the ongoing STS, a national repeated cross-sectional survey of tobacco use in the general population of England. Each month, a new sample of approximately 1700 adults aged ≥16 years is selected using a form of random location sampling. Individuals complete a face-to-face computer-assisted household interview survey with a trained interviewer. The STS samples have been shown to be nationally representative in their sociodemographic composition and proportion of smokers. Full details of the STS methods have been described elsewhere.[23] Ethical approval was granted by the University College London ethics committee.

We used data from 56 monthly waves from May 2012 to December 2016. May 2012 was selected as a starting point, as information on source of tobacco purchase was first measured from this wave. Out of a total of 97 074 respondents, we excluded non-smokers (n=78 184) and respondents with missing smoking status (n=68). Of 18 822 smokers, respondents with missing information on age (n=66) and respondents under 18 years of age (n=259) were excluded, because they could not legally purchase tobacco. We excluded respondents with implausible values for cost of smoking (n=837, assumptions for plausible spending are described below), smokers who did not report their weekly spend on tobacco (n=455) and smokers who did not report their cigarette consumption or reported it to be zero (n=319). We included 16 967 current daily and non-daily cigarette smokers.

### Patient and public involvement

This study involved secondary data analysis of existing data from the STS surveys. Participants and public were not involved in the current study.

### Measurements

The cost of smoking was measured as self-reported weekly spending (in £) on tobacco. Respondents were asked the following open-ended question: 'On average about how much per week do you think you spend on cigarettes or tobacco?'. The cost of smoking was adjusted for inflation using Consumer Prices Index data of all items from the Office for National Statistics,[24] with December 2016 as the reference. Only smokers who adhered to three liberal assumptions of plausible levels of consumption and expenditure per week were included in the analysis, which led to the exclusion of 4.7% of smokers (n=837). The three assumptions included: (1) smokers smoke a maximum of 560 cigarettes per week (n=8), (2) spending does not exceed 280 pounds per week (n=7) and (3) single cigarettes cost between £0.05 and £1 (n=830).

Cost-minimising strategies included (1) reducing consumption of factory-made, (2) reducing consumption of RYO cigarettes (ie, a cheap alternative for factory-made cigarettes[9]), non-UK duty paid tobacco from (3) illicit sources and (4) cross border sources. Factory-made and RYO cigarette consumption were treated as separate continuous variables in all analyses and were expressed in cigarettes per week. Respondents estimated for both factory-made and RYO cigarettes how many cigarettes they smoked per week. For exclusive factory-made cigarette users, RYO consumption levels were zero, and for exclusive RYO users, factory-made cigarette consumption levels were zero.

Purchase from illicit sources was measured as self-reported use of any of the following sources of tobacco at least once in the last 6 months: under the counter (from newsagent, off-license, or corner shop), pub (somebody comes around selling cheap), people who sell cheap cigarettes on the street, people in the local area who are a trusted source of cheap cigarettes or cheap from friends. Cross-border purchasing was measured as self-reported use of cigarettes purchased abroad at least once in the last 6 months. Both were measured as dichotomous variables. Duty free sources within the UK were not specified as a response option and some respondents may have included these in their definition of cross-border sources.

Sociodemographic characteristics measured were gender, age and social grade. Social grade was assigned by the interviewer based on the occupation of the chief income earner of the household and used the National Readership Survey (NRS) classification system to distinguish three categories: *low*: non-working class and (manual) working class (NRS social grades D and E), *middle*: skilled working class and lower middle class (NRS grade C) and *high*: middle class and upper middle class (NRS grades A and B).

Time was measured in months throughout the study period. To control for seasonality (month-of-year effects), the month within the year ('calendar month') was coded as January=1 to December=12.

### Statistical analysis

Data were analysed in R V.3.3.2. The analysis plan was registered on the Open Science Framework prior to data analysis (https://osf.io/ju6tf/). All data and analyses were weighted based on gender, working status, prevalence of children in the household, age, social grade and region, see Fidler *et al*.[23] The use of weighted data was not reported in the analysis plan, but was later decided on to improve the generalisability of the results to the general population of England.

Descriptive statistics are given for the overall sample and stratified by social grade. Trends are graphically described and linear trends were tested using univariate generalised linear models.

Generalised additive models (GAMs) were used to assess the association between cost-minimising strategies and cost of smoking. GAMs are a type of generalised

linear model that allow more sophisticated control for non-linear processes, in this case secular and seasonal trends, than standard linear regression models.[25] The natural log of smoking cost was used, in order to achieve a normal distribution. Results are presented as $100 \cdot \beta$, which in this log-level model can be interpreted as the expected % difference in cost of smoking for a unit increase in the covariate. All models included cyclic cubic regression splines for time (maximum of 5 knots; one for each year) and month within the year (maximum of 12 knots; one for each month). There was no evidence of autocorrelation between time periods according to the autocorrelation function (ACF) and partial ACF, and both the Durbin-Watson test and Breusch-Godfrey test were not statistically significant (p-values, respectively, 0.26 and 0.42).

Model 1 included sociodemographic characteristics (age, gender and social grade). Models 2 to 5 included each cost-minimising strategy separately, adjusting for sociodemographic characteristics. The fully adjusted model (Model 6) included all cost-minimising strategies plus sociodemographics. Model 6 was stratified according to social grade. We assessed effect modification by social grade, by testing interaction between social grade and cost-minimising strategies. Interaction was also tested between cost-minimising strategies and time, in order to assess whether the influence of these strategies on cost changed over time.

We performed two sensitivity analyses using Model 6. First, factory-made cigarette and RYO cigarette consumption were replaced by total cigarette consumption and RYO proportion of that total, as an alternative way of measuring the use of RYO as a substitute for factory-made cigarettes and reflecting its relative cost. Second, the cost per cigarette was used as the outcome instead of cost of smoking per week, as an alternative way of measuring the cost of smoking, that is less dependent on the level of consumption. A posthoc analysis of Model 6 was carried out in the dataset in which we did not exclude values of cost of smoking based on single cigarettes cost (n=17 789).

## RESULTS

Table 1 presents the description of the study population. Factory-made cigarette consumption was lower in the low social grade, while the consumption of RYO tobacco was higher. Illicit sources were more often used in the low social grade, while cross-border purchases were more common in the high social grade. For smokers in the middle social grade, figures for all four price minimising strategies and for reported cost of smoking were in between those of smokers in the low and high social grade.

Figure 1 shows the trends in cost of smoking, consumption and use of illicit and cross-border sources of tobacco. No significant linear trend was found in the cost of smoking (increase of £0.09 per year, 95% CI −0.16 to 0.33, p=0.486). The number of RYO cigarettes consumed per

**Table 1** Weighted description of sociodemographics, cost-minimising strategies and cost of smoking in the overall population of smokers and by social grade

| | | Social grade | | |
|---|---|---|---|---|
| | **Overall population** | **Low (n=7032)** | **Middle (n=8138)** | **High (n=1797)** |
| Age distribution, % (95% CI) | | | | |
| 18 to 24 | 17.2 (16.6 to 17.8) | 18.5 (17.6 to 19.5) | 18.2 (17.3 to 19.1) | 10.4 (9.0 to 11.9) |
| 25 to 34 | 22.1 (21.4 to 22.8) | 23.2 (22.1 to 24.4) | 22.0 (21.0 to 23.0) | 19.4 (17.5 to 21.6) |
| 35 to 44 | 19.4 (18.7 to 20.1) | 18.6 (17.6 to 19.6) | 19.3 (18.3 to 20.2) | 21.8 (19.7 to 24.0) |
| 45 to 54 | 18.6 (18.0 to 19.3) | 17.4 (16.4 to 18.4) | 18.9 (18.0 to 19.9) | 20.7 (18.7 to 22.8) |
| 55 to 64 | 12.5 (12.0 to 13.1) | 12.3 (11.5 to 13.1) | 12.1 (11.4 to 12.9) | 14.6 (13.0 to 16.4) |
| 65+ | 10.2 (9.7 to 10.6) | 10.0 (9.3 to 10.7) | 9.5 (8.9 to 10.1) | 13.1 (11.6 to 14.6) |
| Gender, % (95% CI) | | | | |
| Male | 47.2 (46.4 to 48.0) | 52.1 (5038 to 53.3) | 45.3 (44.2 to 46.5) | 41.7 (39.3 to 44.2) |
| Female | 52.8 (52.0 to 53.6) | 47.9 (46.6 to 49.2) | 54.7 (53.5 to 55.8) | 58.3 (55.8 to 60.7) |
| Social grade, % (95% CI) | | | | |
| Low | 35.7 (34.9 to 36.5) | – | – | – |
| Middle | 50.0 (49.1 to 50.8) | – | – | – |
| High | 14.4 (13.7 to 15.0) | – | – | – |
| Factory-made cigarette consumption in cigarettes/week, mean (95% CI) | 45.9 (45.0 to 46.8) | 43.0 (41.6 to 44.4) | 46.8 (45.5 to 48.1) | 50.1 (47.4 to 52.8) |
| RYO cigarette consumption in cigarettes/week, mean (95% CI) | 35.2 (34.4 to 36.1) | 44.4 (42.9 to 45.9) | 32.4 (31.2 to 33.5) | 22.4 (20.3 to 24.5) |
| Use of illicit sources, % (95% CI) | 8.1 (7.7 to 8.6) | 10.0 (9.2 to 10.8) | 7.7 (7.1 to 8.4) | 5.1 (4.1 to 6.3) |
| Cross-border purchase, % (95% CI) | 8.2 (7.7 to 8.7) | 4.6 (4.1 to 5.2) | 9.2 (8.5 to 9.9) | 13.5 (11.9 to 15.3) |
| Cost of smoking in £/week, mean (95% CI) | 23.3 (23.0 to 23.6) | 22.8 (22.3 to 23.3) | 23.5 (23.1 to 23.9) | 23.9 (22.9 to 24.9) |

RYO, roll-your-own.

week did not significantly change over time (0.32 cigarettes per year, 95% CI −0.36 to 1.00, p=0.346), but there was a significant linear decreasing trend in factory-made cigarette consumption (−1.53 cigarettes per year, 95% CI −2.28 to −0.79, p<0.001). This means that within total cigarette consumption (ie, the sum of RYO and factory-made cigarettes), the proportion of RYO consumption increased (+1.36% per year, 95% CI 0.69 to 2.03, p<0.001, data not shown in figure 1). We found increasing trends in use of illicit sources (+0.53% per year, 95% CI 0.14 to 0.92, p=0.008) and cross-border purchase (+0.41% per year, 95% CI −0.12 to 0.83, p=0.056). A posthoc analysis showed that cross-border purchasing mostly increased in the second half of 2016, but much less up to July 2016 (+0.10 per year, 95% CI −0.36 to 0.55, p=0.664). Trends by social grade are presented in online supplementary figure 1. Linear tests showed that the trends observed in the total population were stronger, and only significant among smokers in the low social grade.

Table 2 presents the associations between cost-minimising strategies and smoking cost. Cost of smoking was higher with increasing age, and higher in the low social grade than in the high social grade. In the fully adjusted model (Model 6), the spline terms for trends over the years (p=0.016) and months (p=0.036) were significant (not presented in table). Model 6 shows that most of the difference between the high and low social grades were attenuated by consumption or source of purchase. When controlling for consumption of factory-made cigarettes in Model 6, a decrease of 10 RYO cigarettes per week was associated with 5.86% lower costs of smoking (95% CI −5.66 to −6.06). Controlling for RYO consumption, a decrease in consumption of 10 factory-made cigarettes was associated with a 12.99% decrease in costs (95% CI −12.80 to −13.18). Both the use of illicit and cross-border sources of tobacco reduced the cost of smoking (illicit: −9.64%, 95% CI −12.94 to −6.33; cross-border: −9.47%, 95% CI −12.74 to −6.20). Due to their low prevalence, use of illicit and cross-border sources combined accounted for only 0.2% of variation in cost of smoking, while factory-made cigarette consumption and RYO consumption accounted for 50% and 10% of the variation in cost of smoking, respectively.

Table 3 shows the associations between cost-minimising strategies and the cost of smoking, by social grade. The same patterns were found in all three groups of social grade, with decreased consumption levels, and use of illicit and cross-border sources all associated with lower cost of

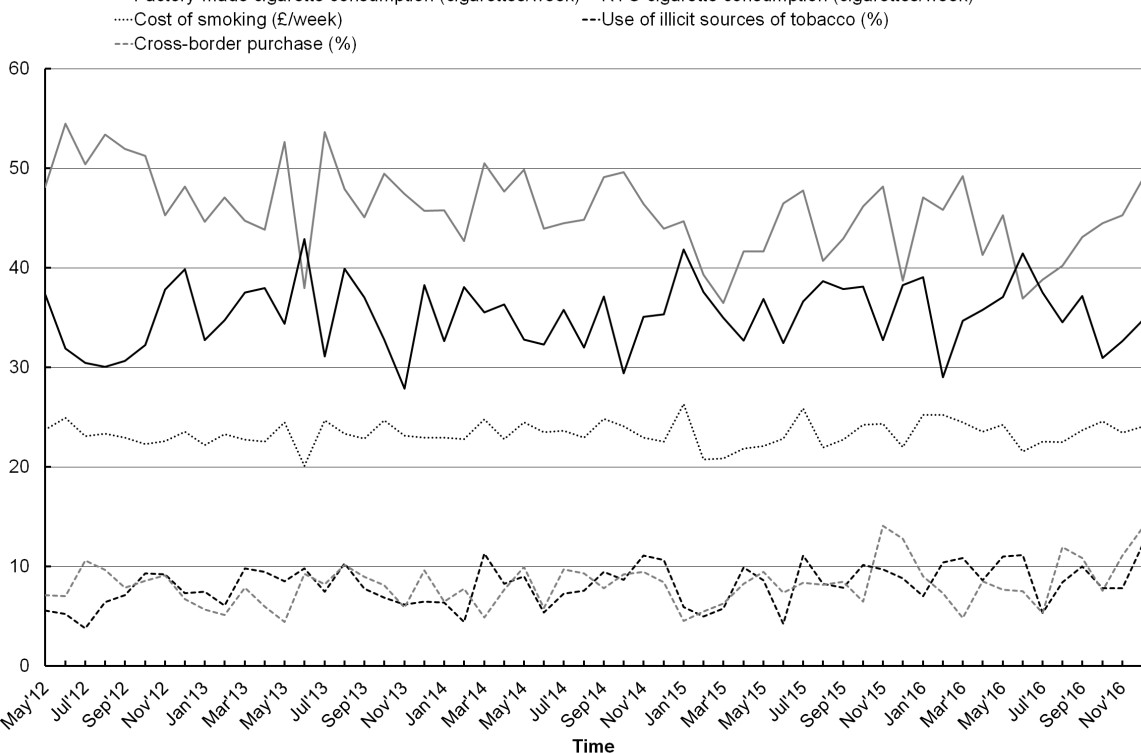

**Figure 1** Weighted trends in cost of smoking, cigarette consumption, use of illicit sources and cross-border purchase among smokers in England. RYO, roll-your-own.

smoking although associations between illicit and cross-border sources of tobacco and cost of smoking tended to be weaker and did not reach statistical significance in the high social grade. Associations of cigarette and RYO consumption with cost tended to be stronger among individuals in the high social grade compared with the low social grade, indicating that they smoke more expensive products.

In table 4, we tested the interaction between time and cost-minimising strategies to assess whether the influence of these strategies on cost changed over time. The association between smoking cost and factory-made cigarette consumption tended to grow stronger over time with −0.12 percentage points per year (95% CI −0.23 to 0.00). Interactions with RYO consumption, use of illicit and cross-border sources of tobacco did not reach statistical significance.

Results for the sensitivity analyses are presented in online supplementary tables. Increasing the proportion of RYO cigarettes within total cigarette consumption (online supplementary table 1) was associated with a decrease in cost of smoking of 70.41% when 0% RYO was compared with 100% RYO (95% CI −72.29 to −68.53). Note that these results were not corrected for the amount of tobacco used for a RYO cigarette versus a manufactured cigarette. The cost per cigarette (online supplementary table 2) increased with a decrease in factory-made and RYO cigarette consumption. It decreased (around 12%) with the use of either illicit or cross-border sources. The posthoc analysis, presented in online supplementary

table 3, demonstrated that the results for the analysis performed on data including individuals who reported very high (>£1) or very low (<£0.05) values for cost per cigarette would not lead to different conclusions than those from the main analysis.

## DISCUSSION

### Key findings

Reported cost of smoking in England did not increase over time above the rate of inflation, despite above-inflation tax increases. Factory-made cigarette consumption decreased, while the proportion of RYO and, to a much lesser extent, illicit and cross-border purchases increased. These trends were only evident in lowest social grade. Lowering factory-made cigarette consumption was associated with greater cost reductions than lowering RYO cigarette consumption. Consumption reduction accounted for 60% of variance. Cross-border and illicit tobacco purchases were associated with lower costs, but due to low prevalence, accounted for only 0.2% of variation in cost. Associations were similar across social grades, although illicit sources reduced the cost more strongly in smokers from low social grades than smokers from high social grades.

### Limitations

The results of this study should be interpreted in light of the following limitations. Use of illicit and cross-border sources was measured dichotomously over a time frame

Table 2  Weighted percentage difference in cost of smoking for sociodemographics and cost-minimising strategies from GAM

| | Percentage difference in cost of smoking (100β with 95% CI) | | |
| --- | --- | --- | --- |
| | Model 1 | Models 2 to 5 | Model 6 |
| | Baseline model | Adjusted for sociodemographics | All variables |
| Age | | | |
| Per 10 years increase | 7.21 (6.42 to 8.01) | | 0.14 (−0.42 to 0.70) |
| Gender | | | |
| Male | ref | | ref |
| Female | 2.22 (−0.34 to 4.78) | | 0.11 (−1.69 to 1.91) |
| Social grade | | | |
| Low | ref | | ref |
| Middle | −1.08 (−3.88 to 1.72) | | 1.44 (−0.52 to 3.40) |
| High | | | −1.50 (−4.33 to 1.32) |
| Factory-made cigarette consumption* | | | |
| Per 10 cigarettes decrease | | −10.17 (−10.35 to −9.99) | −12.99 (−13.18 to −12.80) |
| RYO cigarette consumption† | | | |
| Per 10 cigarettes decrease | | 1.19 (1.44 to 0.94) | −5.86 (−6.06 to −5.66) |
| Use of illicit sources‡ | | | |
| No use of illicit sources in last 6 months | | ref | ref |
| Used illicit sources in last 6 months | | −5.91 (−10.59 to −1.23) | −9.64 (−12.94 to −6.33) |
| Cross-border purchase§ | | | |
| No cross-border purchase in last 6 months | | ref | ref |
| Cross-border purchase in last 6 months | | −7.04 (−11.72 to −2.37) | −9.47 (−12.74 to −6.20) |

*50.4% of variance in spending accounted for by factory-made cigarette consumption.
†9.5% of variance in spending accounted for by RYO cigarette consumption.
‡0.1% of variance in spending accounted for by use of illicit sources of tobacco.
§0.1% of variance in spending accounted for by use of cross-border sources of tobacco.
GAM, generalised additive models; RYO, roll-your-own.

of 6 months preceding the interview. This may have biased the results in two ways. First, we lacked information on the frequency of use of these sources. Because any one-time use is counted as using illicit/cross-border sources and the prevalence may, therefore, not represent, and likely overestimate, the proportion of not full duty-paid purchases out of total tobacco purchases. However, our prevalence rates are comparable with findings from the ITC UK data of 2010/2011 and 2014, in which the source of last purchase was measured.[8 26] Second, the cost of smoking was measured over an average week, a much smaller timeframe than 6 months. Associations between smoking cost and tobacco sources may be diluted as a result of non-differential misclassification and the contribution to the cost of smoking may be larger than portrayed. However, given the very small share of 0.2% according to the current analysis, a substantially large share is unlikely.

The data did not contain information on usual brand. Brand switching is a commonly used cost-minimising strategy.[10 13] Choice of brand may have considerable impact on the cost of smoking, due to undershifting;

the tobacco industry's strategy to divide tax increases disproportionately among different price segments.[9 27] In general, undershifting caused low priced brands to have remained cheap, while prices of premium brands have increased.[9 27 28] The associations found for the studied cost-minimising strategies may still be confounded by brand switching, as brand switching and the studied strategies are likely to co-occur.[10]

This study covers a period in which tobacco taxes increased above inflation annually, but the cost of smoking did not significantly increase above inflation rates. This may reflect a lack of effect of tax increases on the actual retail price of tobacco due to undershifting.[9 27–29] The lack of an increasing trend may, however, also represent selective quitting. Many smokers in England quit during the study period, and smoking prevalence dropped from 19.3% in 2012 to 15.5% in 2016.[19] If smokers who spend more on tobacco are hit harder by a tax increase, they may be more likely to quit in response to increasing tax. The remaining smokers may therefore be those with lower levels of spending to begin with. The current study did not capture any effect of taxes on quitting.

**Table 3** Weighted percentage difference in cost of smoking for sociodemographics and cost-minimising strategies from GAM, stratified by social grade

| | Social grade | | | | | |
| | Low (n=7032) | Middle (n=8138) | | High (n=1797) | |
| | 100β with 95% CI | 100β with 95% CI | P value for interaction, middle vs low | 100β with 95% CI | P value for interaction, high vs low |
|---|---|---|---|---|---|
| **Age** | | | | | |
| Per 10 years increase | −0.23 (−1.04 to 0.59) | −0.18 (−0.65 to 1.00) | | 0.27 (−1.65 to 2.19) | |
| **Gender** | | | | | |
| Male | ref | ref | | ref | |
| Female | 0.54 (2.10 to 3.18) | 0.17 (−2.44 to 2.78) | | −1.55 (−7.56 to 4.46) | |
| **Factory-made cigarette consumption** | | | | | |
| Per 10 cigarettes decrease | −11.70 (−11.41 to −11.98) | −13.20 (−12.93 to −13.47) | <0.001 | −15.36 (−14.74 to −15.98) | <0.001 |
| **RYO cigarette consumption** | | | | | |
| Per 10 cigarettes decrease | −4.95 (−4.68 to −5.21) | −6.15 (−5.84 to −6.45) | 0.007 | −7.90 (−7.10 to −8.70) | 0.004 |
| **Use of illicit sources** | | | | | |
| No use of illicit sources in last 6 months | ref | ref | | ref | |
| Used illicit sources in last 6 months | −12.25 (−16.72 to −7.78) | −7.11 (−12.02 to −2.21) | 0.028 | −5.63 (−19.09 to 7.84) | 0.078 |
| **Cross-border purchase** | | | | | |
| No cross-border purchase in last 6 months | ref | ref | | ref | |
| Cross-border purchase in last 6 months | −13.01 (−19.03 to −6.72) | −9.99 (−14.46 to −5.53) | 0.264 | −7.24 (−15.80 to 1.33) | 0.053 |

All models were adjusted for all variables in the table.
GAM, generalised additive models; RYO, roll-your-own.

We only collected data on expenditure on smoking and not expenditure on alternative nicotine products. In England, e-cigarette use increased over the study period.[30] As a fifth of smokers use e-cigarettes,[31] we have underestimated smokers' expenditure on nicotine. Decreases in tobacco consumption may in part have been due to switching to dual-use of combustible cigarettes and e-cigarettes, and the inverse association found between cigarette consumption and smoking cost would have been somewhat weaker if expenditure on e-cigarettes would have been taken into account.

### Interpretation

Our results are in line with previous findings that RYO cigarettes are much cheaper than factory-made cigarettes[9] as we found that switching from factory-made to RYO cigarettes is an effective cost-mitigating strategy. In our data, the proportion of RYO use increased over time. This increase has been observed since the early 2000s[29] and appears to continue over time.[8] Proportionally increased use is likely to be a response to an increasing gap in prices between factory-made and RYO.[9] Switching to RYO may have serious public health consequences, as smokers

**Table 4** Increase per year in the weighted % difference in cost of smoking for cost-minimising strategies from GAM

| | Increase in the % difference in cost of smoking for each consecutive year (95% CI) | P value for interaction |
|---|---|---|
| Factory-made cigarette consumption · time | −0.12 (−0.23 to 0.00) | 0.051 |
| RYO cigarette consumption · time | 0.00 (−0.12 to 0.13) | 0.992 |
| Illicit sources · time | 0.93 (−1.44 to 3.31) | 0.441 |
| Cross-border purchase · time | −0.23 (−2.54 to 2.09) | 0.848 |

Models were adjusted for age, gender, social grade and other cost-minimising strategies in the table.
GAM, generalised additive models; RYO, roll your-own tobacco.

using RYO tobacco are much less sensitive to further price increases.[32] Moreover, smokers from lower social grades are more likely to use RYO tobacco,[29] which makes the increasing price gap likely to contribute to growing socioeconomic inequalities in smoking.[12] In order to encourage smoking cessation across social grades, taxes on RYO tobacco need to increase to the same level as factory-made cigarettes, which may be achieved through continued larger increases in RYO tobacco taxes.[8 17 33]

We found a small increase in the proportion of smokers who reported purchasing tobacco from illicit or cross-border sources. This is in line with an overview by Rowell *et al*, showing that the most reliable information on illicit tobacco does not show dramatic increases in use.[34] An analysis of 2002–2014 UK ITC data showed no increases in use of self-reported sources outside the UK or from informal sellers.[8] Illicit trade in cigarettes, measured by the tax gap between consumption and sales of tobacco, decreased from 16% in 2005–2006 to 8% in 2014–2015.[22] However, between 2015–2016 and 2016–2017, the tax gap increased to 15%,[22] which this paper reflects. The increase in tax gap in recent years seems mainly due to a decline in consumption rather than a growth in illicit trade.[22] The tax gap for RYO tobacco strongly declined from 60% in 2005 to 28% in 2017.[22]

We found that, in line with previous findings,[8 12 18] individuals from lower social grades were less likely to purchase tobacco abroad and more likely to use illicit sources, than smokers from higher social grades. Although statistical power was limited, associations between consumption levels and costs tended to be stronger among individuals in the high social grade compared with the low social grade. This may be because smokers in lower social grades are more likely to use lower priced brands,[29] for which the reduction of consumption has a smaller effect on the total cost of smoking. The association between illicit sources and the cost of smoking tended to be stronger in smokers from lower social grades, which may be explained by the frequency of using these sources, if smokers from lower social grades use illicit sources on a more regular basis.

### Implications

In order to have actual costs of smoking increase above inflation, this study suggests that tax increases during the period of study were not enough to impact weekly tobacco expenditure. Changes in tobacco taxation policy are required, such as sudden larger tax increases, as called for elsewhere,[35] as these would be more impactful. However, tobacco taxation policies need to be designed in a way that takes industry strategies across brand segments and product types into account.[9 27] As previously called for,[8 9 17 27 29 33] this study makes a strong case for continued higher relative tax increases on RYO tobacco products compared with factory-made cigarettes. Other strategies may include maximising specific taxation, strong minimum price policies, plain packaging that removes price promotions from packs (already in effect in the UK), restricting brands to one variant and preventing the introduction of new brands.[9 27 36 37]

Although illicit tobacco formed only a minor threat to the costliness of smoking, a continued increase is undesirable. Action at the national and international level, including effective implementation of the Framework Convention on Tobacco Control (FCTC) protocol to eliminate illicit trade, has potential to reduce illicit trade in the near future.[38–40]

## CONCLUSION

At a time when tax increases were designed to raise the cost of tobacco 2%–5% above inflation annually, weekly spending on tobacco by smokers in England did not change above inflation. Our study showed that smokers commonly reduce consumption and switch to hand-rolled tobacco, particularly those of more disadvantaged social grades, but that the use of illicit and cross-border sources of tobacco was much less common and hardly contributed to total expenditure on tobacco. Strong future tobacco taxation policies are needed that take industry strategies across brand segments and product types into account.

**Author affiliations**
[1]Department of Public Health, Amsterdam Public Health research institute, Amsterdam UMC, University of Amsterdam, Amsterdam, Netherlands
[2]Department of Behavioural Science and Health, University College London, London, UK
[3]Department of Addictions, Institute of Psychiatry, Psychology and Neuroscience, King's College London, London, UK
[4]Research Department of Clinical, Educational and Health Psychology, University College London, London, UK
[5]Tobacco Control Research Group, Department for Health, University of Bath, Bath, UK

**Contributors** JB, AM and RW conceived and designed the study. MAGK prepared and analysed the data, interpreted the findings and drafted the manuscript. AM, TP, JB, RW, EB and ABG contributed to the interpretation of the data and provided critical revisions to the manuscript. All authors approved the final version of the paper for submission.

**Funding** This work was supported by the National Institute for Health Research Public Health Research (project number 13/43/58). The Smoking Toolkit Study is currently primarily funded by Cancer Research UK (C1417/A22962; C44576/A19501) and has previously also been funded by Pfizer, GSK and the Department of Health.

**Disclaimer** Department of Health Disclaimer: The views and opinions expressed therein are those of the authors and do not necessarily reflect those of the Public Health Research programme, NIHR, NHS or the Department of Health. No funders had any involvement in the design of the study, the analysis or interpretation of the data, the writing of the report or the decision to submit the paper for publication.

**Competing interests** RW undertakes consultancy and research for and receives travel funds and hospitality from manufacturers of smoking cessation medications. EB and JB have received unrestricted research funding from Pfizer.

**Patient consent for publication** Not required.

**Provenance and peer review** Not commissioned; externally peer reviewed.

**Data sharing statement** For access to the Smoking Toolkit Study, please contact Dr Jamie Brown, jamie.brown@ucl.ac.uk.

and indication of whether changes were made. See: https://creativecommons.org/licenses/by/4.0/.

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
