## [Reviewer comments · BMJ Open]

ARTICLE DETAILS

TITLE (PROVISIONAL)	Smokers' strategies across social grades to minimise the cost of smoking in a period with annual tax increases: Evidence from a national survey in England
AUTHORS	Kuipers, Mirte; Partos, Timea; McNeill, Ann; Beard, Emma; Gilmore, Anna; West, Robert; Brown, Jamie

VERSION 1 - REVIEW

REVIEWER	Kelvin Choi National Institute on Minority Health and Health Disparities Division of Intramural Research
REVIEW RETURNED	25-Sep-2018

GENERAL COMMENTS	The authors used the monthly STS data to assess the effect of tax increases on trends in cigarette cost minimizing behaviors by social grade. This is an important line of research and I applaud the authors for a nicely written manuscript. I have the following comments/suggestions: 1. Introduction: Please provide the month of the tax increases between 2012 and 2016. Given the monthly data use, it will be beneficial for the readers to know.2. Introduction: The authors may want to provide other price-related regulations in the UK, e.g., prohibition on price promotions, in-store discounts, etc.3. Sample: Although the exclusion criteria make sense in general, I recommend performing sensitivity analysis by including those excluded from the analyses presented. One particular concern I have is excluding low and high single cigarette cost. Some smokers may get cigarettes from friends, the cost of some cigarettes could be close to zero. This is particularly common among young adult smokers who are non-daily smokers. At the same time, if singles are sold in the UK (legally or illegally), cost of cigarettes may exceed the specified limit. Since the analytic approach used is sensitive to outliers, I think the additional sensitivity analysis will show the impact of the exclusion criteria on the findings.4. Statistical analysis: The data were unweighted. Given the sampling frame, it seems to be natural that the analysis would be weighted. Please provide a rationale. Also, since the data is unweighted, using univariate generalized linear model is inadequate to assess trends since the sample differences over time can confound the association. At the minimum, demographics should be controlled for.
--

	5. Statistical analysis: Did the authors examine potential non-linear trends? It seems to warrant the exploration given unequal tax increases over time. 6. Statistical analysis: As strong argument of tax increases is that they promote smoking cessation. Can the authors provide the trends in current smoking during the observation period? It will provide a better backdrop for the readers to place the findings. 7. Statistical analysis: It seems like it would be of interest to perform the analysis to match the timing of different amount of tax increases. E.g., the high tax increase for RYO cigarettes may have driven the increase in cross-boarder purchase in the second half of 2016. 8. Implications: The data from this study do not seem to support large tax increases. They seem to suggest that despite repeated tax increases, smokers who chose to continue smoking are able to maintain a steady cigarette expenditure. To deferentially impact low social grade smokers, other interventions such as higher RYO tax increases with enforcement on cross-border purchases may be warrants. Other non-tax approaches discussed by Golden (https://tobaccocontrol.bmj.com/content/25/4/377) may also be helpful.
--	--

REVIEWER	Filippos Filippidis Imperial College London, UK
REVIEW RETURNED	27-Sep-2018

GENERAL COMMENTS	Abstract. There is no indication what the beta coefficients refer to. A sentence in the methods or a mention of the unit in the abstract would help to quantify the effect estimated in the study. Introduction, 2nd paragraph. I have no concern regarding the statements in this paragraph, but I think it would be useful to make some distinction between countries/regions. These strategies are not necessary applicable to all settings, so I encourage the authors to mention where these studies have been conducted, at least whether they were in the UK or elsewhere. Introduction, 3rd paragraph. I think it's not clear if the tax increase in 2012 was 5% in total or 5% above inflation. Similarly for the other increases mentioned there. Please rephrase to clarify. Methods/Measurements. I am a bit sceptical about the assumption, particularly the one regarding cost per cigarette. While the condition itself is very reasonable, it leads to the exclusion of many observations. I can think of many reasons why someone might fall outside these limits. The most obvious could be that people do not know exactly how many cigarettes they smoke, but I think they might be likely to remember how many packs they bought in a week. In any case, could the authors discuss this in more detail, as it excludes about 5% of the observations and it could introduce bias? At least describing what kind of responses the excluded individuals gave and comparing them with the sample analysed. Methods/Measurements. Can the authors cite evidence that RYO is indeed cheaper than factor-made cigarettes in the UK? This is not
--

	necessarily the case in every single country. Methods/Measurements. I think one more sentence is needed to explain how respondents were classified into social grade groups. Is there a question, a series of questions? Self-reported, combination of variables? This analysis only included smokers. It would be useful to know how much the prevalence of smoking changed in England during this period; a major change might have implications for the interpretation of results. Discussion. The prevalence of e-cigarette use increased quite a bit during the study period. Many are dual users (cigarettes and e-cigarettes), which might have major implications for the analysis and the conclusions of the study. It could be that, for dual users, cost of smoking has not increased, but the cost of their nicotine addiction has, because they spent money on e-cigarettes as well. I think this element should not be ignored. Regardless of whether one believes that e-cigarettes are good or bad for public health, they would be expected to play a role in how smokers respond to price changes, especially in the UK. Also, I am not sure I agree with the authors' view that tax increases have failed to increase cost of smoking. Sure, higher increases would be great, but if those tax increases have led those who continue to smoke to reduce consumption, they have had at least some success. And this ignores the number of smokers who managed to quit overall. I would suggest them to consider this point.
--	--

REVIEWER	Abraham Brown Nottingham Trent University United Kingdom
REVIEW RETURNED	29-Sep-2018

GENERAL COMMENTS	This is a timely study that uses nationally representative sample of smokers across social grade to compare the contribution of price minimising strategies to the cost of smoking. Evidence suggests that such price minimising behaviours can decrease public health benefits that are gained from increasing cigarette prices through taxation. Indeed, the tobacco industry, knowing that most smokers especially from high-income countries engage in price minimising behaviours, have utilised these strategies to market their products. However, I have a few concerns. The authors found that switching from factory-made to RYO cigarettes is an effective cost-mitigating strategy. However, it is unfortunate the Smoking Toolkit Study does not include data on usual brand. This is because smokers have higher propensity to purchase cheaper brands or switch to lower priced brands in the face of tax increases. It is therefore worrying that this study did not assess smokers' propensity to switch brands. Although they acknowledged that brand switching and the strategies examined are likely to co-occur, its exclusion weakens the study as evidence suggests that smokers can be price sensitive irrespective of income. As switching is strongly related to price increases, one can't ascertain whether the findings are truly reflective of smokers' cost-minimisation behaviours. Aside from this, perhaps, it would have been revealing to examine association between increase in tobacco taxes and quitting
---

	behaviours. Studies have shown that tax increases and avoidance are associated with cost-minimisation behaviours and quitting behaviours. Another concern is the use of a repeated cross-sectional data. If indeed the aim of this study is to assess trends, then a longitudinal study design would have been the most appropriate approach to use. The benefits of a longitudinal analysis over a repeated cross-sectional study include increased statistical power and the ability to estimate a greater range of conditional probabilities. Another weakness is the fact that unweighted sample was used. This means that less ability to make appropriate inferences regarding changes in strata proportions with time. As such estimates of the model that was stratified by social grade may be less accurate because the data was unweighted. The authors noted that linear trends were examined using generalised linear models but this method could have been used for non-linear trends as well. It is also unclear which trends were linear and those that were non-linear. It might help to explain the rationale for sensitivity analysis. In the results section, page 7, paragraph 1, if the number of RYO cigarettes consumed per week did not significantly change over time, how do the authors conclude that the proportion of RYO within total cigarette consumption increased? Is it possible that the significant linear decreasing trend in factory-made cigarette consumption was a consequence of smokers switching to cheaper priced brands? Lastly, although expected, the authors should exercise caution in discussing reported purchasing of tobacco from illicit or cross-border sources because this lacked statistical power and the variance explained was almost negligible.
--	---

VERSION 1 – AUTHOR RESPONSE

Reviewer: 1

Reviewer Name: Kelvin Choi

Institution and Country: National Institute on Minority Health and Health Disparities Division of Intramural Research

Please state any competing interests or state 'None declared': None declared.

Please leave your comments for the authors below

The authors used the monthly STS data to assess the effect of tax increases on trends in cigarette cost minimizing behaviors by social grade. This is an important line of research and I applaud the authors for a nicely written manuscript. I have the following comments/suggestions:

1. Introduction: Please provide the month of the tax increases between 2012 and 2016. Given the monthly data use, it will be beneficial for the readers to know.

Response: The tax increases occurred in March of each year, which we had stated in the introduction: "In 2012-2016, taxes on all tobacco products increased above inflation nationwide in March of each year".

2. Introduction: The authors may want to provide other price-related regulations in the UK, e.g., prohibition on price promotions, in-store discounts, etc.

Response: In the UK, legislation on price promotions and discounts did not change across the study period until the Tobacco Products Directive, including standardised packaging (which prohibited discounts and price reductions), came into force in May 2017. The transition period began in May 2016 but retailers only began substantially adhering in the last months of this period, i.e during 2017.(1) As the results of the current study will therefore not be influenced by recent restriction on price promotions and discounts, we decided not to add this information to the manuscript.

3. Sample: Although the exclusion criteria make sense in general, I recommend performing sensitivity analysis by including those excluded from the analyses presented. One particular concern I have is excluding low and high single cigarette cost. Some smokers may get cigarettes from friends, the cost of some cigarettes could be close to zero. This is particularly common among young adult smokers who are non-daily smokers. At the same time, if singles are sold in the UK (legally or illegally), cost of cigarettes may exceed the specified limit. Since the analytic approach used is sensitive to outliers, I think the additional sensitivity analysis will show the impact of the exclusion criteria on the findings.

Response: We have performed an additional analysis on data including a wider definition of plausible values of weekly spend on smoking. We only excluded those smoking more than 560 cigarettes per week (N=8), and those spending more than 280 pounds per week (N=7). The total population was 17,789. Results were to those in the main analysis, and would not lead to different conclusion. The results are presented in Supplementary Table 3. We now report at the end of the methods section that this post-hoc analysis was undertaken: "A post-hoc analysis of Model 6 was carried out in the dataset in which we did not exclude values of cost of smoking based on single cigarettes cost (N=17,789)." We now also report at the end of the results section that "The post-hoc analysis, presented in Supplementary Table 3, demonstrated that the results for the analysis performed on data including individuals who reported very high (>£1) or very low (<£0.05) values for price per cigarette would not lead to different conclusions than those from the main analysis."

4. Statistical analysis: The data were unweighted. Given the sampling frame, it seems to be natural that the analysis would be weighted. Please provide a rationale. Also, since the data is unweighted, using univariate generalized linear model is inadequate to assess trends since the sample differences over time can confound the association. At the minimum, demographics should be controlled for.

Response: The GAM analyses were always adjusted for sociodemographics. We have repeated the trend analysis on weighted data, for which a figure is added at the end of this document. Trends remain very similar to the presented results. Only cross-border purchase showed a smaller change of +0.41 per year (95%CI -0.01 to 0.83) compared with +0.52% per year (95%CI 0.17 to 0.88) in the original analysis. As this is a minor difference, with a large overlap in confidence intervals, we decided to maintain our analysis, as we had pre-specified it (<https://osf.io/ju6tf/>), in the paper.

In the methods we added: "Weighted analyses were performed as a post-hoc sensitivity analysis, and resulted in similar findings that would lead to the same conclusions."

5. Statistical analysis: Did the authors examine potential non-linear trends? It seems to warrant the exploration given unequal tax increases over time.

Response: Trend analyses were intended to be descriptive (see aim: to describe trends in in the cost of smoking and use of cost-minimising strategies between 2012 and 2016). The linear trend was quantified to provide an overall estimate of the change over the years, but the exact development over time is described in the figure.

In the GAM analysis we do control for non-linear trends over the years, and within years, using splines. These splines, however, do not provide readily interpretable quantifications of trends. In the 3rd paragraph of the results, we now report the following: "In the fully adjusted model (Model 6), the

spline term for trend over the years was significant ($p=0.007$), while the trend within years was not ($p=0.233$; not presented in table).”

6. Statistical analysis: A strong argument of tax increases is that they promote smoking cessation. Can the authors provide the trends in current smoking during the observation period? It will provide a better backdrop for the readers to place the findings.

Response: As the analysis is restricted to smokers, we believe it would be confusing to add smoking prevalence to the presented results in the figures as this would suggest that the data would include non-smokers. Instead we have added information on the decrease in smoking prevalence in the introduction (3rd paragraph): “where there has been a marked decrease in smoking prevalence (from 19.3% in 2012 to 15.5% in 2016(19))”; and discussion section, where the limitation of selected quitting is discussed: “Many smokers in England quit during the study period, and smoking prevalence dropped from 19.3% in 2012 to 15.5% in 2016.(19)”

7. Statistical analysis: It seems like it would be of interest to perform the analysis to match the timing of different amount of tax increases. E.g., the high tax increase for RYO cigarettes may have driven the increase in cross-border purchase in the second half of 2016.

Response: Although interesting, we consider this type of analysis outside the scope of the current study as we did not aim to quantify the association between tax increases and cost of smoking. Moreover, a test of trends between financial years would not provide sufficient evidence to causally attribute differences in trends to differences in taxation given the limited number of years and variation in the level of tax increases.

8. Implications: The data from this study do not seem to support large tax increases. They seem to suggest that despite repeated tax increases, smokers who chose to continue smoking are able to maintain a steady cigarette expenditure. To differentially impact low social grade smokers, other interventions such as higher RYO tax increases with enforcement on cross-border purchases may be warrants. Other non-tax approaches discussed by Golden (<https://tobaccocontrol.bmj.com/content/25/4/377>) may also be helpful.

Response: Thank you for this comment.

As stated in our previous response, we did not set out to assess responses to different tax increases but overall our study indicates that the tobacco tax changes implemented were not enough to have an impact. We therefore do believe that larger tax increases, and unplanned ones, as suggested elsewhere, would be more impactful. We have modified the first sentence of the implications section to clarify this:

“In order to have actual costs of smoking increase above inflation, this study suggests that tax increases during the period of study were not enough to impact weekly tobacco expenditure. Changes in tobacco taxation policy are required, such as sudden larger tax increases, as called for elsewhere,(35) as these would be more impactful.”

As this reviewer suggests however, we had then gone on to say, that higher RYO tax increases would be particularly helpful: “However, tobacco taxation policies need to be designed in a way that takes industry strategies across brand segments and product types into account.(9, 27). As previously called for, (8, 9, 17, 27, 29, 33) this study makes a strong case for continued higher relative tax increases on RYO tobacco products compared with factory-made cigarettes.” We had also referred to the effective implementation of the FCTC protocol to capture the reviewer’s point about enforcement of cross-border purchases: “Action at the national and international level, including effective

implementation of the FCTC protocol to eliminate illicit trade, has potential to reduce illicit trade in the near future.(37-39)”

The Golden et al. review argues that price promotions and minimum price policies in particular would add to existing tax-based policies. We list these, together with other measures, in which we now specify that plain packaging removes price promotions from packs: “Other strategies may include maximising specific taxation, strong minimum price policies, plain packaging that removes price promotions from packs (already in effect in the UK), restricting brands to one variant and preventing the introduction of new brands.(9, 27, 36, 37)”. We added Golden et al. as a reference.

Golden et al. also discuss price caps to avoid low priced cigarette brand varieties, We are hesitant in suggesting price caps, because this may lower the overall prices (as was explained in the review).

Reviewer: 2

Reviewer Name: Filippos Filippidis

Institution and Country: Imperial College London, UK

Please state any competing interests or state ‘None declared’: None declared

Please leave your comments for the authors below

Thank you for the opportunity to review this interesting study. The topic is important and the data used appropriate. I have some concerns regarding aspects that were not considered, but this is overall well written. Detailed comments below:

Abstract. There is no indication what the beta coefficients refer to. A sentence in the methods or a mention of the unit in the abstract would help to quantify the effect estimated in the study.

Response: We reworded the results in the abstract to clarify the meaning of the coefficients, e.g.: “Cost of smoking was 12.65% lower with consumption of 10 fewer factory-made cigarettes (95%CI=-12.84;-12.46)”.

Introduction, 2nd paragraph. I have no concern regarding the statements in this paragraph, but I think it would be useful to make some distinction between countries/regions. These strategies are not necessary applicable to all settings, so I encourage the authors to mention where these studies have been conducted, at least whether they were in the UK or elsewhere.

Response: As studies are from different countries, often combining multiple countries (in ITC study papers), indicating each country would likely make the paragraph too crowded. Instead we indicated that evidence is from high-income countries, and in the next paragraph we specify which studies include at least some data from the UK: “Studies across high-income countries estimate that between half and three quarters of all smokers apply cost-minimising strategies” and “The current study was set in the UK, where there has been a marked decrease in smoking prevalence (from 19.3% in 2012 to 15.5% in 2016(19)) and where some use of cost-minimising strategies has been demonstrated using data from the International Tobacco Control policy evaluation project.(8, 11, 12, 18)”

Introduction, 3rd paragraph. I think it's not clear if the tax increase in 2012 was 5% in total or 5% above inflation. Similarly for the other increases mentioned there. Please rephrase to clarify.

Response: We now specify that all tax increases were above inflation: "In 2012-2016, taxes on all tobacco products increased nationwide in March of each year, by 5% above inflation in 2012, and 2% above inflation in 2013-2015. In 2016, a differential tax of 5% above inflation for RYO tobacco and 2% above inflation for factory-made cigarettes was applied".

Methods/Measurements. I am a bit sceptical about the assumption, particularly the one regarding cost per cigarette. While the condition itself is very reasonable, it leads to the exclusion of many observations. I can think of many reasons why someone might fall outside these limits. The most obvious could be that people do not know exactly how many cigarettes they smoke, but I think they might be likely to remember how many packs they bought in a week. In any case, could the authors discuss this in more detail, as it excludes about 5% of the observations and it could introduce bias? At least describing what kind of responses the excluded individuals gave and comparing them with the sample analysed.

Response: We have performed an additional analysis on data including a wider definition of plausible values of weekly spend on smoking. We only excluded those smoking more than 560 cigarettes per week (N=8), and those spending more than 280 pounds per week (N=7). The total population was 17,789. Results were to those in the main analysis, and would not lead to different conclusion. The results are presented in Supplementary Table 3. We now report at the end of the methods section that this post-hoc analysis was undertaken: "A post-hoc analysis of Model 6 was carried out in the dataset in which we did not exclude values of cost of smoking based on single cigarettes cost (N=17,789)." We now also report at the end of the results section that "The post-hoc analysis, presented in Supplementary Table 3, demonstrated that the results for the analysis performed on data including individuals who reported very high (>£1) or very low (<£0.05) values for price per cigarette would not lead to different conclusions than those from the main analysis."

Methods/Measurements. Can the authors cite evidence that RYO is indeed cheaper than factory-made cigarettes in the UK? This is not necessarily the case in every single country.

Response: A reference was added to Hiscock et al.(2): "reducing consumption of RYO cigarettes (i.e. a cheap alternative for factory-made cigarettes (9))"

Methods/Measurements. I think one more sentence is needed to explain how respondents were classified into social grade groups. Is there a question, a series of questions? Self-reported, combination of variables?

Response: We now clarify this in the text: "Social grade was assigned by the interviewer based on the occupation of the chief income earner of the household and used the National Readership Survey classification system to distinguish three categories"

This analysis only included smokers. It would be useful to know how much the prevalence of smoking changed in England during this period; a major change might have implications for the interpretation of results.

Response: We have added information on the change in smoking in the limitations section, where selective quitting is discussed: "Many smokers in England quit during the study period, and smoking prevalence dropped from 19.3% in 2012 to 15.5% in 2016.(19)"

Discussion. The prevalence of e-cigarette use increased quite a bit during the study period. Many are dual users (cigarettes and e-cigarettes), which might have major implications for the analysis and the conclusions of the study. It could be that, for dual users, cost of smoking has not increased, but the cost of their nicotine addiction has, because they spent money on e-cigarettes as well. I think this element should not be ignored. Regardless of whether one believes that e-cigarettes are good or bad

for public health, they would be expected to play a role in how smokers respond to price changes, especially in the UK.

Response: This paper is specifically focussed on cost of tobacco smoking, and our analysis of those particular costs have not been compromised by expenditure on e-cigarette use. However, we do agree that combustible tobacco does not capture the full extent of expenditure on nicotine. Unfortunately our data do not contain information on expenditure on alternative nicotine products, and we are therefore unable to take this into account in our analysis. We now do acknowledge in the limitations section that particularly for reductions in cigarette consumption, results may be affected:

“We only collected data on expenditure on smoking, and not expenditure on alternative nicotine products. In England, e-cigarette use increased over the study period.(30) As a fifth of smokers use e-cigarettes,(31) we have underestimated smokers’ expenditure on nicotine. Decreases in tobacco consumption may in part have been due to switching to dual-use of combustible cigarettes and e-cigarettes, and the inverse association found between cigarette consumption and smoking cost would have been somewhat weaker if expenditure on e-cigarettes would have been taken into account.”

Also, I am not sure I agree with the authors’ view that tax increases have failed to increase cost of smoking. Sure, higher increases would be great, but if those tax increases have led those who continue to smoke to reduce consumption, they have had at least some success. And this ignores the number of smokers who managed to quit overall. I would suggest them to consider this point.

Response: We now acknowledge more explicitly in the limitations section that we do not capture the effects that taxes have on quitting: “The current study did not capture any effect of taxes on quitting.” (see page 7)

Reviewer: 3

Reviewer Name: Abraham Brown

Institution and Country: Nottingham Trent University, United Kingdom

Please state any competing interests or state ‘None declared’: None declared

Please leave your comments for the authors below

This is a timely study that uses nationally representative sample of smokers across social grade to compare the contribution of price minimising strategies to the cost of smoking. Evidence suggests that such price minimising behaviours can decrease public health benefits that are gained from increasing cigarette prices through taxation. Indeed, the tobacco industry, knowing that most smokers especially from high-income countries engage in price minimising behaviours, have utilised these strategies to market their products. However, I have a few concerns.

The authors found that switching from factory-made to RYO cigarettes is an effective cost-mitigating strategy. However, it is unfortunate the Smoking Toolkit Study does not include data on usual brand. This is because smokers have higher propensity to purchase cheaper brands or switch to lower priced brands in the face of tax increases. It is therefore worrying that this study did not assess smokers’ propensity to switch brands. Although they acknowledged that brand switching and the strategies examined are likely to co-occur, its exclusion weakens the study as evidence suggests that smokers can be price sensitive irrespective of income. As switching is strongly related to price increases, one can’t ascertain whether the findings are truly reflective of smokers’ cost-minimisation behaviours.

Response: Thank you for raising this concern. We agree that the paper would have been stronger had we been able to include brand information, to additionally study brand switching as a cost-mitigating strategy. We had acknowledged this in the second paragraph of the limitations section and in the bullet point limitations upfront. However, the current analysis is still valuable to understand the role of the cost-mitigating strategies that we were able to study. In particular, the contribution of use of illicit and cross-border sources to the cost of smoking and the differences between social grades had not been previously quantified. Even though the associations for illicit and cross-border sources may be confounded by potentially co-occurring brand switching, and may therefore be overestimated, their contribution to the cost of smoking was found to be negligible.

Aside from this, perhaps, it would have been revealing to examine association between increase in tobacco taxes and quitting behaviours. Studies have shown that tax increases and avoidance are associated with cost-minimisation behaviours and quitting behaviours.

Response: Smokers' quitting behaviours lay outside the scope of the current study. Unravelling the complex interrelations between tax increases, quitting behaviours, cost of smoking, and cost-mitigating behaviours as an alternative strategy would require a very different type of analysis, and this was not the aim of the current study.

Another concern is the use of a repeated cross-sectional data. If indeed the aim of this study is to assess trends, then a longitudinal study design would have been the most appropriate approach to use. The benefits of a longitudinal analysis over a repeated cross-sectional study include increased statistical power and the ability to estimate a greater range of conditional probabilities.

Response: Population trends are very well measured in repeat cross-sectional data. In longitudinal data, especially over a longer period of time, the sample would develop into a more and more selective group over time of smokers who did not quit. The Smoking Toolkit Study data shows the situation among a new representative sample of smokers at the time of data collection.

We do agree that behaviours such as switching to RYO from factory-made cigarettes are best studied longitudinally, as this occurs over time, but unfortunately we did not have such data available.

Another weakness is the fact that unweighted sample was used. This means that less ability to make appropriate inferences regarding changes in strata proportions with time. As such estimates of the model that was stratified by social grade may be less accurate because the data was unweighted.

Response: We performed an additional analysis on the weighted data. We repeated the fully adjusted model in the total population, and the stratified analysis by social grade. Results are presented in the table at the end of this document, and demonstrate that weighting would not change the conclusions that were drawn from the original analysis. We report this post-hoc analysis in the methods section: "Weighted analyses were performed as a post-hoc sensitivity analysis, and resulted in similar findings that would lead to the same conclusions."

The authors noted that linear trends were examined using generalised linear models but this method could have been used for non-linear trends as well. It is also unclear which trends were linear and those that were non-linear.

Response: Trend analyses were intended to be descriptive (see aim: to describe trends in the cost of smoking and use of cost-minimising strategies between 2012 and 2016). The linear trend was quantified to provide an overall estimate of the change over the years, but the exact development over time is described in the figure.

It might help to explain the rationale for sensitivity analysis.

Response: RYO consumption can be measured as an absolute (number of RYO cigarettes) or relative measure (RYO as a proportion of total consumption). We decided a-priori to analyse the absolute measure, because smokers who only use RYO can still cut back as a cost-mitigating strategy without increasing their proportion of RYO relative to factory-made. When studying cigarette consumption as a cost-mitigating strategy, the interpretation of the absolute measure is therefore the same for all types of smokers (i.e., only factory-made or RYO cigarettes, or a mix of both). Still, the proportion of RYO shows us the use of RYO as an alternative to factory-made cigarettes and more clearly illustrates that using RYO tobacco is much cheaper than using factory-made cigarettes, and we therefore considered this sensitivity analysis interesting for the reader. We added an explanation to the statistical analysis section: "First, factory-made cigarette and RYO cigarette consumption were replaced by total cigarette consumption and RYO proportion of that total, as an alternative way of measuring the use of RYO as a substitute for factory-made cigarettes and reflecting its relative cost."

The cost of smoking can be measured as an absolute amount per week (in pounds), but can also be measured relative to the number of cigarettes smoked (price per cigarette). The a-priori focus of the paper is on the total cost. However, price per cigarette is less dependent of consumption levels, and this sensitivity analysis may be helpful to the interpretation for some readers. We added an explanation to the statistical analysis section: "Second, the cost per cigarette was used as the outcome instead of cost of smoking per week, as an alternative way of measuring the cost of smoking, that is less dependent on the level of consumption."

In the results section, page 7, paragraph 1, if the number of RYO cigarettes consumed per week did not significantly change over time, how do the authors conclude that the proportion of RYO within total cigarette consumption increased?

Response: We rephrased the sentence to clarify that total consumption is the sum of factory-made and RYO consumption, which logically means that the proportion of RYO increases if the absolute consumption of factory-made decreases and while RYO consumption stays about the same: "This means that within total cigarette consumption (i.e., the sum of RYO and factory-made cigarettes), the proportion of RYO consumption increased (+0.75% per year, 95%CI 0.13 to 1.38, p=0.020, data not shown in Figure 1)."

Is it possible that the significant linear decreasing trend in factory-made cigarette consumption was a consequence of smokers switching to cheaper priced brands?

Response: We consider it unlikely that those who switch brands will smoke fewer cigarettes as a consequence. The opposite may actually be the case if those who switch to cheaper brands are able to continue smoking the same number of cigarettes after an increase in price, while those who do not switch may need to cut down. This could however not be measured in the current data due to lack of information on brand smoked and we therefore avoid speculation on this issue.

Lastly, although expected, the authors should exercise caution in discussing reported purchasing of tobacco from illicit or cross-border sources because this lacked statistical power and the variance explained was almost negligible.

Response: We acknowledge the negligible contribution of use of illicit and cross-border sources to the cost of smoking in our key findings, implications and conclusion. We therefore consider the current text to be cautious in the interpretation of our findings, but we are open to specific suggestions for improvements of parts of the text on this point.

In the results section we now provide the full results including confidence intervals, to show that there is some imprecision in estimating these associations: "Both the use of illicit and cross-border sources

of tobacco reduced the cost of smoking (Illicit: -10.17%, 95%CI: -13.46 to -6.88; cross-border: -10.64%, 95%CI: -14.04 to -7.25).”

VERSION 2 – REVIEW

REVIEWER	Filippos Filippidis Imperial College London, UK
REVIEW RETURNED	31-Jan-2019

GENERAL COMMENTS	My comments have been addressed; I have no further comments.
--

REVIEWER	Kelvin Choi National Institute on Minority Health and Health Disparities, USA
REVIEW RETURNED	05-Feb-2019

GENERAL COMMENTS	Thanks for responding my concerns previously raised. I think the authors have been responsive to my suggestions. One thing I would strongly encourage the authors doing is to present weighted analyses findings instead of unweighted findings. While authors argued that the results are similar, the weighted findings are considered generalizable. This will allow authors and other researchers to cite the findings for the UK as a whole, instead of sample-specific unweighted findings.
---

VERSION 2 – AUTHOR RESPONSE

We would like to thank the reviewers for reading the revised manuscript.

We have revised all tables and figures to show the weighted results. In the text, we have made minor changes to reflect the small changes in the results.

VERSION 3 – REVIEW

REVIEWER	Kelvin Choi National Institute on Minority Health and Health Disparities, USA
REVIEW RETURNED	06-May-2019

GENERAL COMMENTS	Thank you for presented the weighted results.
---